# DRIP: Invariance-preserving Data Reduction for Domain Generalization

## Abstract

Domain Generalization (DG) aims to enable deep models trained on several source domains to generalize to unseen target domains. Existing works assume that data contain both invariant features, whose relationship with label is invariant across domains, and spurious features, which are spuriously correlated to the label. It is widely recognized that retaining invariant features while suppressing spurious ones is critical to achieving DG. However, despite this inspiration, current methods often struggle to guarantee a separation between these features, or their guarantees need relatively strong assumptions of data. In this work, we propose **DRIP** (**D**ata **R**eduction with **I**nvariance-**P**reserving), a data augmentation paradigm theoretically proven to improve domain generalization. We prove that such DRIP can reduce spurious features while preserving invariant features, forcing the model to rely more on generalizable invariant features during training. Following the principles of DRIP, we propose several low cost implementations and verify that one of those indeed meets the criteria of DRIP. Experiments show that it consistently outperforms existing DG baseline in two modalities, vision and IMU.

## 1 Introduction

Deep learning has demonstrated remarkable performance when the test data share the same or a similar distribution as the training data. However, in real-world scenarios, it is difficult to guarantee this distributional assumption, which substantially degrades the performance of deep models. Domain generalization (DG) formulates this challenge as a domain shift problem and seeks to enable models trained on a limited set of source domains to generalize effectively to unseen target domains. Causality-based approaches (Arjovsky et al., 2019; Ahuja et al., 2021; Li et al., 2022; Vuong et al., 2025) impose further constraints on this problem. They assume that the data contain latent invariant features, which are also named as causal feature and causally decide the label, as well as spurious features, which are spuriously correlated to the label. Algorithms are designed to learn representations that capture former while minimizing the influence of latter, thereby yielding an invariant predictor. Data augmentation approaches (Nguyen et al., 2023; Xu et al., 2021; Zhou et al., 2021; 2020; Li et al., 2021) are typically confined to the visual modality, where domain shift is identified as specific factors such as style or texture. These methods mitigate the impact of domain shift by applying style transfer or texture-disrupting transformations.

However, although causality-based methods are designed around the invariance principle, they rarely provide theoretical guarantees for separating invariant features from spurious ones (Li et al., 2022; Rosenfeld et al., 2021). In particular, when the diversity of training domains is insufficient, it is difficult to avoid the influence of pseudo-invariant features (see Subsection 3.2). In addition, some of these methods require domain labels for the data (Arjovsky et al., 2019; Ahuja et al., 2021), which introduces extra annotation costs in practical settings. Data augmentation methods are inspired by human prior knowledge, but they also lack theoretical guarantees. Moreover, empirical evidence has shown that existing DG methods often fail to consistently outperform ERM (Gulrajani & Lopez-Paz, 2021). Ruan et al. (2022) attributes this empirical observation to the lack of discriminative information about the target domain. They argue that textual descriptions of images can fill this gap, since such descriptions remain invariant regardless of whether they originate from the source or the target domain. While this approach improves DG performance, it relies on vision–language models (e.g., CLIP (Radford et al., 2021)) or large-scale image–text paired data, which is impractical for other modalities and for many other vision tasks.

To address these issues, we take a step back and seek a low-cost yet theoretically grounded data augmentation strategy that can preserve invariant features while reducing spurious ones. We theoretically prove that if a transformation $R$ (1) compresses the information contained in the input data, while (2) preserving the information necessary for classification, it can reduce the influence of spurious features while maintaining the invariant ones. We refer to the transformations that satisfy these two conditions as **DRIP** (**D**ata **R**eduction with **I**nvariance-**P**reserving). Intuitively, DRIP can be regarded as a do-like intervention on the feature space (though not in the strict causal sense). By its second defining requirement, $R$ cannot alter the invariant features, since they are causally related to the label $Y$. Therefore, $R$ can only serve to weaken the spurious features. We provide a more rigorous proof later, which brings a solid theoretical guarantee.

We stress that DRIP is a theoretical notion, which does not prescribe how to implement such an $R$ in practice. The key, however, is that the implementation must be carefully designed or selected to satisfy the requirements, particularly the second one, which demands that it does not compromise the discriminative information necessary for the label. In this sense, choosing an appropriate $R$ essentially introduces knowledge about the target domain in the form of an inductive bias: that is, $R$ should not change the label of a sample, whether it comes from the source or the target domain. This objective is highly similar to that of Ruan et al. (2022), but can be achieved at a much lower computational cost. Besides, such knowledge about the target domain makes DRIP more promising to solve pseudo-invariant issue.

Beyond theoretically proving that DRIP can improve DG, we also identified several candidate transformations that may satisfy the definition, and analyzed whether they truly meet it and lead to improved performance. Since the theoretical guarantee of DRIP is general, its applicability is not confined to a specific modality. For the visual modality and IMU-based HAR (inertial-measurement-units-based human activity recognition), we introduce three versions of dropout directly applied to the input data. In addition, we propose two methods tailored specifically for the IMU modality. Experiments validate their effectiveness and further demonstrate the theoretical value of DRIP. Our contributions are as follows:

- We propose the theoretical concept of DRIP, and prove that it can preserve causal features while reducing spurious ones, which have been widely recognized as beneficial for improving DG.
- We propose an implementation of DRIP, namely applying channel-wise dropout directly to the data. It shows improvement over existing baselines and outperforms other implementations that do not fully meet the definition. For comparison, we also provide other implementations that do not fully meet the definition.
- We present, to the best of our knowledge, the first theoretically grounded data augmentation that demonstrates effectiveness across multiple modalities. Empirical evaluations confirm its superior performance on both visual and IMU datasets.

## 2 RELATED WORKS

### 2.1 DOMAIN GENERALIZATION

Domain Generalization (DG) aims to learning representations that can generalize well on unseen domains. Our approach can be categorized both as a causality-based approach and as a data augmentation approach. **Causality-based** approaches formalize regularization terms guided by causal theory. IRM (Arjovsky et al., 2019) encourages the model to achieve optimality across all domains by using the derivative of the loss within each domain as a penalty term. Some other works encourage domain invariance by increasing the similarity of risks (Krueger et al., 2021; Nguyen et al., 2024) or features (Chevalley et al., 2022) across domains. Some (Ahuja et al., 2021; Li et al., 2022) build upon IRM and integrate information bottleneck (IB) (Tishby et al., 2000) principle to compress spurious features. Though inspired by the domain-invariance principle, these methods seldom provide qualitative or quantitative analyses of how well they separate causal features from spurious ones (Rosenfeld et al., 2021; Li et al., 2022; Krueger et al., 2021). As shown below in the analysis in Subsection 3.2, such designs fail to clearly remove spurious features. In contrast, we prove that DRIP can preserve invariant features while reducing spurious ones, and we establish a lower bound of spurious feature reduction. Moreover, our approach does not require domain labels for the data.

**Data augmentation** approaches address this from the perspective of the pattern of data itself, aiming to weaken style or texture information in images. Some (Zhou et al., 2020; Zhang et al., 2022; Robey et al., 2021; Nguyen et al., 2023; Yang et al., 2021) use additional models to generate new data.Some works randomly mix data from different domains, either in the data space (Mancini et al., 2020; Yue et al., 2019; Yan et al., 2020; Jeon et al., 2021; Hendrycks et al., 2019) or in the hidden space (Zhou et al., 2021; Li et al., 2021). Some (Carlucci et al., 2019; Xu et al., 2021) apply independent transformations to each image. These rely on human priors and typically require additional computational cost. In contrast, DRIP is theoretically grounded, and requires minimal additional computational cost.

There are also algorithms based on meta-learning (Qiao & Peng, 2021; Bui et al., 2021; Shu et al., 2021), ensemble learning (Lee et al., 2022; Chen et al., 2024) and disentangled representation learning (Bai et al., 2021; Nam et al., 2021; Zhang et al., 2022; Wang et al., 2021). However, empirical evidence shows that existing algorithms struggle to consistently improve upon ERM (Gulrajani & Lopez-Paz, 2021; Chen et al., 2024). Ruan et al. (2022) attributes it to the lack of discriminative information about the target domain. It argues that textual descriptions of images are invariant across domains, including the target domains, and therefore leverages image–text paired data to address the problem. However, in many real-world scenarios, massive textual description data are unavailable, which limits the practicality of this approach. The core idea of DRIP is similar, namely to identify a data transformation that remains invariant across domains. This eliminates the need for textual data and make DRIP more feasible in real scenarios.

## 2.2 Data Augmentation beyond DG Scenarios

Beyond DG, data augmentation has been widely used to reduce overfitting and improve model performance. In addition to common techniques such as random cropping and flipping (Krizhevsky et al., 2012; He et al., 2016), there are also more targeted methods. For example, some works (DeVries & Taylor, 2017; Singh et al., 2018; Zhong et al., 2020; Hoyer et al., 2023) introduce random masking to force the model to capture more global features. These works are very similar to the patch-wise version of DRIP. In contrast, our method is derived from theoretical analysis, and we further demonstrate that the patch-wise version is not the optimal choice for DG tasks.

## 3 Method

### 3.1 Preliminaries

We define domain generalization (DG) as training a model on data from multiple source domains, such that it can generalize to unseen target domains at test time. Specifically, for each domain (also named as environment) $e \in \mathcal{E}$, the input variable $X^e \in \mathcal{X}$ and the label variable $Y^e \in \mathcal{Y}$ are sampled from a joint distribution $(X^e, Y^e) \sim \mathbb{P}^e$. DG requires using the data $\mathcal{E}_{\text{tr}} = \{(x_i^e, y_i^e) : e \in \mathcal{E}_{\text{tr}}, 1 \le i \le n^e\}$ from training domains $\mathcal{E}_{\text{tr}} \subsetneq \mathcal{E}$ to find a optimal model $f$ on unseen domains $\mathcal{E} \backslash \mathcal{E}_{\text{tr}}$:

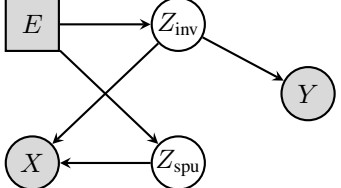

$$\min_f \frac{1}{|\mathcal{E} \backslash \mathcal{E}_{\text{tr}}|} \sum_{e \in \mathcal{E} \backslash \mathcal{E}_{\text{tr}}} R^e(f) \quad (1)$$

Figure 1: A causal graph describing data generation process. Observed variables are shaded.

where $R^e(f) = \mathbb{E}_{(x^e, y^e) \sim \mathbb{P}^e}[\ell(f(x^e), y^e)]$ and $\ell$ is the loss function. Empirical Risk Minimization (ERM) (Vapnik, 1999) is a common algorithm to train $f$. Its training objective is to minimize the expected risk over all training domains, which is expressed as:

$$\mathcal{L}_{\text{ERM}} = \frac{1}{|\mathcal{E}_{\text{tr}}|} \sum_{e \in \mathcal{E}_{\text{tr}}} R^e(f) \quad (2)$$

However, without additional restrictions on $\mathcal{E}$, the objective in Equation 1 may lack a meaningful solution. For example, in binary classification, if the difference between domains is so dramatic that the optimal model for $\mathcal{E}_{\text{tr}}$ is $f$, while the optimal model for $\mathcal{E} \backslash \mathcal{E}_{\text{tr}}$ is exactly $1 - f$, then no satisfactory algorithm can be expected to solve the problem. Therefore, some assumptions are necessary. We

consider using a causal graph to describe the problem (see Figure 1). In this graph, we use $E$ to denote the domain and two latent features $Z_{\text{inv}}$ and $Z_{\text{spu}}$ to denote the key features that generate the input $X$ (Li et al., 2022; Vuong et al., 2025). $Z_{\text{inv}}$ is the causal feature that determines the label $Y$ and $Z_{\text{spu}}$ is the environmental feature that *spuriously* correlated with $Y$. For example, in a photo of a horse, the horse itself corresponds to $Z_{\text{inv}}$, while the background (e.g. a grass field) corresponds to $Z_{\text{spu}}$, due to their spurious correlation. Even if the background is replaced with a desert, the label remains 'horse', indicating that $Z_{\text{spu}}$ is spuriously correlated with $Y$ but not causally related.

Due to $E \perp\!\!\!\perp Y \mid Z_{\text{inv}}$, $Z_{\text{inv}}$ is also referred to the *invariant* feature, indicating that $P(Y^e \mid Z^e_{\text{inv}})$ is invariant for all $e \in \mathcal{E}$. This suggests that if we can find a representation that maps the input $X$ to $Z_{\text{inv}}$ while discarding $Z_{\text{spu}}$, this representation can generalize to unseen domains.

### 3.2 Failure on Pseudo-invariant Features

The idea of preserving invariant features and reducing spurious features has inspired numerous methods, such as IRM, to design their own regularization terms. These methods are intuitively reasonable, as a representation that preserves only invariant features is indeed one solution to their training objective. However, there might exist deteriorating solutions. The root cause is the presence of pseudo-invariant features, which are essentially spurious features but exhibit invariant behavior on insufficient training domains. Roughly speaking, an environment can at most distinguish one spurious feature (Li et al., 2022). When the number of training domains is small or the diversity is insufficient, pseudo-invariant features can arise and damage performance on unseen domains. Since pseudo-invariant features share the same statistical properties as invariant features in the training data, methods that rely solely on the invariance principle are inherently unable to distinguish between the two, leading to the leak of pseudo-invariant features. Li et al. (2022) recognizes this issue and attempts to use the information bottleneck principle to force the model to learn the minimal set of features while ensuring invariance, with the goal of excluding pseudo-invariant features. However, because these two feature are not distinguishable in the training domains, it may still end up learning a solution that includes both pseudo-invariant features and partial invariant features. We give an example here to demonstrate how invariance and IB principles fails:

**Example 1.** *(Pseudo-invariant Features) Suppose training domains include $\mathcal{E}_{tr} = \{1, 2\}$ and target domain includes $\mathcal{E}\backslash\mathcal{E}_{tr} = \{3\}$. The causal model is defined as:*

$$
\begin{aligned}
Z^{p,e=1}_{spu} = Z^{s,e=1}_{spu} &= Z^{e=1}_{inv} \sim Bernoulli(0.8) \\
Z^{p,e=2}_{spu} = 1 - Z^{s,e=2}_{spu} &= Z^{e=2}_{inv} \sim Bernoulli(0.2) \\
1 - Z^{p,e=3}_{spu} = Z^{s,e=3}_{spu} &= Z^{e=3}_{inv} \sim Bernoulli(0.8) \\
Z_{spu} = (Z^p_{spu}, Z^s_{spu}), \quad X = (Z_{inv}, Z_{spu}&), \quad Y = Z_{inv} \oplus N
\end{aligned}
$$

*where $N \sim Bernoulli(0.1)$ is a binary noise, $N \perp\!\!\!\perp (Z_{inv}, Z_{spu})$ and $\oplus$ is XOR operation.*

In terms of $\mathcal{E} = \{1, 2, 3\}$, both $Z^{\text{p}}_{\text{spu}}$ and $Z^{\text{s}}_{\text{spu}}$ are spurious features. However, with the source domains $\mathcal{E}_{\text{tr}}$, $Z^{\text{p}}_{\text{spu}}$ shows invariant behavior, which make it a pseudo-invariant feature. Even with the invariance and IB principles, both $Z^{\text{p}}_{\text{spu}}$ and $Z_{\text{inv}}$ are empirically optimal solutions. Detailed discussion is in Subsection A.1. Only from a more global perspective (or with information about the target domain) can the two be distinguished.

The analysis above indicates that when the training data is insufficient and the source domains are limited (which is quite common), additional information about the target domain are needed. This is very similar to the conclusion of Ruan et al. (2022). Ruan et al. (2022) introduces an inductive bias based on textual descriptions, assuming that the textual description of an image remains invariant in the target domain. We aim to introduce information about the target domain in a more lightweight manner, without requiring additional data, by finding a data augmentation method that does not alter the labels even in the target domain. This leads to the following rigorous definition of DRIP.

### 3.3 Theoretical foundations of DRIP

Now we propose a data augmentation method at the theoretical level and prove that it can improve DG. We define DRIP as follows:

**Definition 2.** *(DRIP) Define an operator $R : \mathcal{X} \to \mathcal{X}$ if it satisfies both conditions:*

$$P(Y \mid R(X)) = P(Y \mid X), \tag{A}$$

$$H(R(X)) < H(X). \tag{B}$$

The operator $R$, which reduces irrelevant information from $X^e$ while preserving all information necessary for predicting $Y^e$, can be seen as a form of masking or dropout operation. We will discuss how to choose a specific $R$ in the next subsection. Before presenting our theoretical conclusions, we first introduce a necessary assumption about the data generation process:

**Assumption 3.** *(Generation Process of $X$) $X = c(Z_{inv}, Z_{spu}, N_x)$, where $N_x$ denotes an independent noise and $c$ is a deterministic and invertible function.*

It assumes that the information of $Z_{\text{inv}}$ and $Z_{\text{spu}}$ can be recovered from $X$. This is reasonable because we do not concern with information that is unrelated to the observable variables. This assumption has been adopted by many studies (Arjovsky et al., 2019; Rosenfeld et al., 2021; Ahuja et al., 2021; Sun et al., 2020). Building on the above, we derive the following proposition:

**Theorem 4.** $I(X; Z_{inv}) = I(R(X); Z_{inv})$.

This shows that the transformation $R$ does not reduce the causal features. Intuitively, $R$ can be loosely understood as an operator similar to the do-like intervention, though not in the strict sense. According to the causal graph in Figure 1, there is a causal edge $Z_{\text{inv}}^e \to Y^e$. If $R$ were to modify $Z_{\text{inv}}$, the conditional distribution of $Y$ would change. This contradicts Condition A in Definition 2. Thus, we obtain an intuitive proof by contradiction for Theorem 4. We also have:

**Theorem 5.** $I(X; Z_{spu}, N_x) - I(R(X); Z_{spu}, N_x) \geq H(X) - H(R(X)) \geq 0$.

Theorem 4 and Theorem 5 show that DRIP reduces the influence of spurious features while preserving invariant features. This aligns with the objective and rationale of causality based approaches and provide a straightforward guarantee of performance. More importantly, the requirements of DRIP do not focus on latent features but instead directly address the properties of observed variables. This makes it easier to verify whether a method satisfies the requirements and also yields a simpler and more effective theoretical result. By selecting a simple yet appropriate $R$, invariance can be enforced from a more global perspective (i.e., with respect to $\mathcal{E}$ rather than only $\mathcal{E}_{\text{tr}}$), thereby introducing discriminative information about the target domain. This makes it more promising than prior approaches in addressing the issue of pseudo-invariant features. The formal proof of Theorem 4 and 5 is provided in Subsection A.2 and A.3.

In addition to the theoretical derivations above, we also provide a practical criterion for determining whether a transformation satisfies Definition 2. For Condition B, it is relatively easy for humans to directly assess whether a transformation reduces the amount of information. Condition A, however, is more ambiguous. Beyond direct assessment by human experts, we propose a empirical criterion for verification. If we assume a sufficiently powerful model that can capture the relationship between $X$ and $Y$, then the converged value of its training loss can serve as an quantitative indicator of this relationship. We will fully discuss this criterion in Subsection A.4 and apply it in Subsection 5.1. When applying it, we extend the number of training epochs and repeat multiple trials to approximate the idealized model assumption as closely as possible.

**Criterion 6** (Informal)**.** *Assume we have a sufficiently powerful model. If the final converged value of the training loss with $R$ is similar to that without $R$, we consider it an empirical evidence of $R$ to satisfy Condition A.*

### 3.4 IMPLEMENTATIONS OF DRIP

We have mathematically shown that any $R$ satisfying Definition 2 preserves the information in $Z_{\text{inv}}$ while reducing the influence of $Z_{\text{spu}}$, which is expected to improve domain generalization performance. We now turn to the discussion of its concrete implementation. Since $R$ is an operator on $\mathcal{X}$ that reduces the information in $X$, a natural idea is to design it as a random dropout operation. We consider three potential candidates, namely channel-wise, patch-wise, and pixel-wise. The semantic information of an image is usually reflected in the shape of objects (Geirhos et al., 2019). Channel-wise dropout keeps the shape of the image, but patch-wise dropout may cover the target

Table 1: Results on DomainBed. Best results are in **bold**. The suffixes (*Ch.*, *Pa.*, and *Pi.*) of DRIP correspond to the channel-wise, patch-wise, and pixel-wise versions, respectively. Channel-wise DRIP shows notable improvements over ERM and other baselines.

| Algorithm | VLCS | PACS | OfficeHome | TerraIncognita | DomainNet | Avg |
|---|---|---|---|---|---|---|
| ERM | $77.5 \pm 0.4$ | $85.5 \pm 0.2$ | $66.5 \pm 0.3$ | $46.1 \pm 1.8$ | $40.9 \pm 0.1$ | 63.3 |
| IRM | $78.5 \pm 0.5$ | $83.5 \pm 0.8$ | $64.3 \pm 2.2$ | $47.6 \pm 0.8$ | $33.9 \pm 2.8$ | 61.6 |
| GroupDRO | $76.7 \pm 0.6$ | $84.4 \pm 0.8$ | $66.0 \pm 0.7$ | $43.2 \pm 1.1$ | $33.3 \pm 0.2$ | 60.9 |
| CORAL | $\mathbf{78.8 \pm 0.6}$ | $86.2 \pm 0.3$ | $68.7 \pm 0.3$ | $47.6 \pm 1.0$ | $41.5 \pm 0.1$ | 64.6 |
| Mixup | $77.4 \pm 0.6$ | $84.6 \pm 0.6$ | $68.1 \pm 0.3$ | $47.9 \pm 0.8$ | $39.2 \pm 0.1$ | 63.4 |
| DANN | $78.6 \pm 0.4$ | $83.6 \pm 0.4$ | $65.9 \pm 0.6$ | $46.7 \pm 0.5$ | $38.3 \pm 0.1$ | 62.6 |
| CDANN | $77.5 \pm 0.1$ | $82.6 \pm 0.9$ | $65.8 \pm 1.3$ | $45.8 \pm 1.6$ | $38.3 \pm 0.3$ | 62.0 |
| SagNet | $77.8 \pm 0.5$ | $86.3 \pm 0.2$ | $68.1 \pm 0.1$ | $48.6 \pm 1.0$ | $40.3 \pm 0.1$ | 64.2 |
| Fish | $77.8 \pm 0.3$ | $85.5 \pm 0.3$ | $68.6 \pm 0.4$ | $45.1 \pm 1.3$ | $42.7 \pm 0.2$ | 64.0 |
| Fishr | $77.8 \pm 0.1$ | $85.5 \pm 0.4$ | $67.8 \pm 0.1$ | $47.4 \pm 1.6$ | $41.7 \pm 0.0$ | 64.0 |
| VREx | $78.3 \pm 0.2$ | $84.9 \pm 0.6$ | $66.4 \pm 0.6$ | $46.4 \pm 0.6$ | $33.6 \pm 2.9$ | 61.9 |
| IIB | $77.2 \pm 1.6$ | $83.9 \pm 0.2$ | $68.6 \pm 0.1$ | $45.8 \pm 1.4$ | $41.5 \pm 2.3$ | 63.4 |
| RSC | $77.1 \pm 0.5$ | $85.2 \pm 0.9$ | $65.5 \pm 0.9$ | $46.6 \pm 1.0$ | $38.9 \pm 0.5$ | 62.7 |
| MLDG | $77.2 \pm 0.4$ | $84.9 \pm 1.0$ | $66.8 \pm 0.6$ | $47.7 \pm 0.9$ | $41.2 \pm 0.1$ | 63.6 |
| EQRM | $77.8 \pm 0.6$ | $\mathbf{86.5 \pm 0.2}$ | $67.5 \pm 0.1$ | $47.8 \pm 0.6$ | $41.0 \pm 0.3$ | 64.1 |
| DRIP (Ch.) | $78.7 \pm 0.2$ | $85.0 \pm 0.1$ | $\mathbf{71.7 \pm 0.1}$ | $\mathbf{48.7 \pm 0.6}$ | $\mathbf{45.6 \pm 0.2}$ | $\mathbf{65.9}$ |
| DRIP (Pa.) | $78.3 \pm 0.7$ | $84.6 \pm 0.4$ | $71.1 \pm 0.3$ | $44.2 \pm 1.4$ | $44.6 \pm 0.1$ | 64.6 |
| DRIP (Pi.) | $74.7 \pm 0.4$ | $83.8 \pm 0.4$ | $66.7 \pm 0.3$ | $29.1 \pm 3.3$ | $42.0 \pm 0.0$ | 59.3 |

object. In more severe cases, if the target occupies only a small portion of the image, it may be completely covered, leading to a total loss of class information. Subsection 5.1 provides experimental validation of these conclusions based on Criterion 6. Pixel-wise dropout, to some extent, introduces an additional task of restoring the original image from one that contains pixel noises. This indicates that such dropout cannot reliably reduce the information in the data. On the contrary, it introduces an irrelevant objective that interferes with model training, which in turn harms model performance. In summary, applying channel-wise dropout directly to the data should effectively improve DG performance, while patch-wise and pixel-wise version might show inferior performance on DG tasks.

Beyond the three primary candidates, we also introduce two additional implementations on IMU datasets. The first one treat the three axes of each sensor as a joint signal and then either randomly project it onto a two-dimensional plane with a random orientation (denoted as *Proj*) or replace it with the magnitude of the signal (denoted as *Norm*). The second one constructs a VAE with a small latent space and a shallow CNN architecture (denoted as *VAE*), which is frozen after pretrained and used for generating compressed data. These are relatively complex and less transferable to other modalities. Nevertheless, they are inspired by the concept of DRIP and demonstrate improvements over ERM (see Subsection 5.5). This indicates that the theoretical value of DRIP is not limited to specific implementations, further reinforcing our theoretical contribution.

## 4 EXPERIMENTS

### 4.1 DOMAINBED RESULTS

#### 4.1.1 EXPERIMENTAL SETUP

For visual modality, we conduct experiments on 5 datasets, which are PACS (Li et al., 2017), VLCS (Fang et al., 2013), OfficeHome (Venkateswara et al., 2017), TerraIncognita (Beery et al., 2018) and DomainNet (Peng et al., 2019). We implement our method based on DomainBed (Gulrajani & Lopez-Paz, 2021), using a ResNet-50 (He et al., 2016) pretrained on ImageNet (Deng et al., 2009) as the backbone. The specific dropout rate in DRIP is sampled from $\{0.1, 0.2, 0.3\}$, and the patch size of patch-wise dropout is uniformly sampled from $\{4, 8, 16, 32\}$. Other hyperparameters

Table 2: Results on IMU datasets. Macro F1 and accuracy are reported. Best results are in **bold**. The suffixes (*Ch.*, *Pa.*, and *Pi.*) of DRIP correspond to the channel-wise, patch-wise, and pixel-wise versions, respectively. Channel-wise DRIP shows notable improvements over ERM, while patch-wise and pixel-wise version perform similar to ERM.

| | DSADS | | OPPORTUNITY | | PAMAP2 | | MOTIONSENSE | | HAPT | | Avg | |
|---|---|---|---|---|---|---|---|---|---|---|---|---|
| | mF1 | Acc | mF1 | Acc | mF1 | Acc | mF1 | Acc | mF1 | Acc | mF1 | Acc |
| ERM | 83.41 | 85.72 | 56.54 | 85.25 | 80.28 | 84.30 | **93.17** | **94.13** | 84.90 | 93.99 | 79.66 | 88.68 |
| DRIP (Ch.) | **88.87** | **90.50** | **67.51** | **88.51** | **84.99** | **86.64** | **93.17** | 93.80 | 85.04 | 93.63 | **83.92** | **90.62** |
| DRIP (Pa.) | 82.76 | 85.00 | 59.37 | 85.86 | 81.82 | 84.77 | 92.74 | 93.72 | **85.40** | 93.94 | 80.42 | 88.66 |
| DRIP (Pi.) | 82.70 | 85.42 | 57.09 | 85.33 | 82.31 | 85.34 | 92.60 | 93.68 | 84.53 | **94.13** | 79.85 | 88.78 |

(e.g., batch size, dropout, etc.) are randomly sampled following the settings in DomainBed. Each test domain is evaluated with 3 random seeds each with 20 trials, and training-domain validation was used for model selection.

### 4.1.2 RESULTS

Results are listed in Table 1. Channel-wise version of DRIP outperforms all baselines, whereas pixel-wise version performs poorly. Channel-wise version yields consistent improvements across most datasets, with performance on PACS being comparable to baselines. Patch-wise one also demonstrates strong performance on most datasets but shows a substantial disadvantage on TerraIncognita. This is because, in TerraIncognita, the target animals occupy only a small portion of the image. According to the previous analysis, patch-wise dropout is poorly suited for such datasets.

## 4.2 IMU HAR RESULTS

### 4.2.1 EXPERIMENTAL SETUP

We also conduct experiments on IMU datasets, which are DSADS (Barshan & Yüksek, 2014), OPPORTUNITY (Chavarriaga et al., 2013), PAMAP2 (Reiss & Stricker, 2012), MOTION-SENSE (Malekzadeh et al., 2019; 2020) and HAPT (Reyes-Ortiz et al., 2016). Results are based on a ResNet-like CNN, and the models are trained from scratch. The dropout rate in DRIP is 0.2 and the patch size in patch-wise version of DRIP is 64. All experiments are conducted in leave-one-user-out setting, where each user is considered as a domain. Details of datasets and other settings are shown in Section B.

### 4.2.2 RESULTS

Results are listed in Table 2. Channel-wise version of DRIP outperforms ERM while other two versions show comparable performance. On MOTIONSENSE and HAPT, the results of channel-wise version are similar to those of ERM. The reason may be that these two datasets contain a larger number of users (24 and 30), which leads to a greater number of training domains. Compared with the other three datasets (8, 4, and 8), having more training domains allows the model to more easily learn invariant features, thereby reducing the additional gains brought by DRIP. However, this does not diminish the contribution of DRIP, since in many real-world scenarios it is often difficult to collect data from a sufficiently large number of domains (Li et al., 2017; Fang et al., 2013). We further analyze the influence of the number of training domains in Subsection 5.4.

## 5 ANALYSIS

### 5.1 VALIDATION OF WHETHER IT MEETS THE DEFINITION

Since the theoretical guarantee of improvement comes from the conditions in Definition 2, it is therefore crucial to ensure that a specific implementation satisfies it. Verifying Condition B is trivial, so we focus on how to verify Condition A. Based on the proposed Criterion 6, we examine whether the training loss convergence values of the channel-wise and patch-wise versions of DRIP are similar

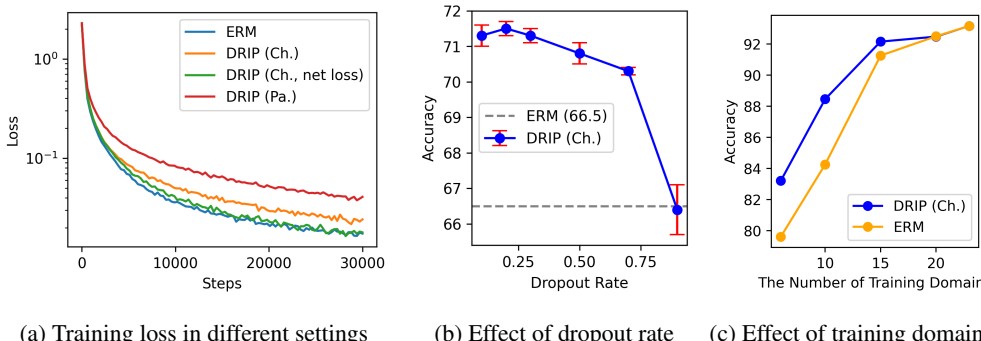

(a) Training loss in different settings    (b) Effect of dropout rate    (c) Effect of training domains

Figure 2: (a) The training loss curve of ERM, channel-wise DRIP and patch-wise DRIP. The definition of *net loss* is explained in Equation 3. The net loss converges to a similar value to that of ERM. (b) Performance of different dropout rate in DRIP. A dropout rate around 0.2 shows optimal performance and dropout rates up to 0.7 consistently show improvements over ERM. (c) Comparison between DRIP and ERM as the number of training domains increases. When the number of training domains is small, the improvement of DRIP is evident. As the number of training domains increases, the improvement gradually diminishes.

to those of ERM. Experiments are conducted on TerraIncognita. To examine the final convergence value, we extended the training steps to 30k. These training steps are irrelevant to the main experiments and are used solely to verify the information about the label $Y$ contained in the data. For the channel-wise version, since the data have only three RGB channels, there is a non-negligible probability that the entire input will be erased. For example, when the dropout rate $p$ is 0.2, this probability is $0.2^3 = 0.008$. To enable accurate verification, we further introduce the concept of *net loss*:

$$\text{loss}_{\text{net}} = (\text{loss} - \hat{H} \times p^C)/(1 - p^C) \tag{3}$$

where $\hat{H} = \log(M)$ is the cross-entropy between the model output and the label when the data are completely erased. Assuming that the output distribution when the input is completely erased resembles the marginal distribution $P(Y)$, its average cross-entropy with the label is $\hat{H} = H(Y)$. We count the samples in each class and computed $\hat{H}$ and the net loss of the channel-wise version. Since the probability of complete erasure for the patch-wise version is very low (e.g. $0.2^{(224/32) \times (224/32)} \approx 5.63 \times 10^{-35}$), we do not take its net loss into account. As shown in Figure 2a, the channel-wise version (with net loss) converges to a value similar to that of ERM. In contrast, the convergence value of the patch-wise version is significantly larger, which is consistent with the analysis in Subsection 4.2.

## 5.2 THE HYPERPARAMETER

Compared with ERM, channel-wise dropout introduces only one additional hyperparameter, the dropout rate. In the DomainBed experiments, it was uniformly sampled from $\{0.1, 0.2, 0.3\}$. To further investigate the effect of this hyperparameter, we evaluated the performance when the dropout rate was fixed at specific values. The experiments were conducted on OfficeHome and we report results with dropout of 0.1, 0.2, 0.3, 0.5, 0.7 and 0.9. As shown in Figure 2b, when the dropout rate falls within an appropriate range, DRIP exhibits consistent performance improvements. This also confirms that a dropout rate around 0.2 is a reasonable setting.

## 5.3 RESULTS IN I.I.D. SCENARIOS

To complement our contribution, we also evaluate DRIP in i.i.d. setting. The purpose here is to verify that DRIP improves performance in DG scenarios while not degrading performance on data that share the same domains as the training data. Experiments are conducted on the IMU datasets under within-user setting. In this setting, the data from all users are mixed and then randomly split into training and test sets, ensuring that both sets come from the same distribution. As shown in

Table 3: Results in i.i.d. scenarios. Experiments are conducted on IMU datasets. DRIP (ch.) denotes the channel-wise version of DRIP. DRIP does not show degraded performance in the i.i.d. setting. This indicates that DRIP is not a trade-off method that sacrifices i.i.d. performance.

| | DSADS | | OPPORTUNITY | | PAMAP2 | | MOTIONSENSE | | HAPT | | Avg | |
|---|---|---|---|---|---|---|---|---|---|---|---|---|
| | mF1 | Acc | mF1 | Acc | mF1 | Acc | mF1 | Acc | mF1 | Acc | mF1 | Acc |
| ERM | 78.90 | 81.60 | 58.60 | 80.95 | 93.17 | 92.98 | 96.70 | 96.98 | 86.85 | 96.03 | 82.84 | 89.71 |
| DRIP (ch.) | 86.86 | 87.07 | 59.80 | 89.54 | 93.15 | 93.35 | 96.99 | 97.02 | 86.04 | 94.35 | 84.57 | 92.27 |

Table 3, DRIP performs on par with ERM under this setting. This indicates that the improvements in DG settings are not at the cost of degraded i.i.d performance, ensuring the robustness of DRIP.

## 5.4 PERFORMANCE WHEN THE NUMBER OF TRAINING DOMAINS VARIES

As analyzed in Subsection 4.2, the improvement of DRIP diminishes as the number of training domains increases. Here, we investigate the impact of the number of training domains on the performance of DRIP. We conducted experiments on the MOTIONSENSE dataset. Based on the leave-one-user-out setting, only a subset of the training domains was used in each test domain. We then compared the performance of the channel-wise version of DRIP with that of ERM. As shown in Figure 2c, when the number of training domains is relatively small (fewer than about 15), DRIP outperforms ERM. As the number of training domains increases, the performance of the two methods converges. Since DRIP improves DG by reducing spurious features, having more training domains enables the model to better distinguish spurious features on its own, which gradually reduces the benefit of DRIP.

## 5.5 PERFORMANCE OF OTHER IMPLEMENTATIONS

Figure 3: Results of three additional implementations.

| | DSADS | | OPPORTUNITY | | PAMAP2 | |
|---|---|---|---|---|---|---|
| | mF1 | Acc | mF1 | Acc | mF1 | Acc |
| ERM | 83.41 | 85.72 | 56.54 | 85.25 | 80.28 | 84.30 |
| DRIP (Ch.) | 88.87 | 90.50 | 67.51 | 88.51 | 84.99 | 86.64 |
| DRIP (Norm) | 87.88 | 88.99 | 59.05 | 86.31 | 83.78 | 85.91 |
| DRIP (VAE) | 87.83 | 89.24 | 58.15 | 85.65 | 82.15 | 84.91 |

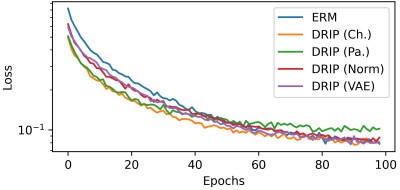

Figure 4: Training loss on IMU task.

We evaluate the other two implementations that designed only for IMU data. These methods are relatively complex to implement, computationally expensive, and almost only effective for sensor-based data. Here, we include them merely to illustrate that the theoretical value of DRIP extends beyond the dropout-based implementations. In Table 3, they show improvements over ERM, although they still fall short of our primary implementation. We also include the training loss curves of these methods in Figure 4 as the Criterion 6 evaluation. It shows that both *VAE* and *Norm* satisfy our Criterion.

## 6 CONCLUSION

We propose a theoretical data augmentation concept, DRIP, for the domain generalization setting and prove that transformations strictly satisfying its definition reduce spurious features while preserving invariant ones. This offers a simple yet principled approach with the potential to address the issue of pseudo-invariant features. We further instantiate DRIP through several implementations in both visual and IMU modalities, achieving consistent improvements on the corresponding datasets. In particular, applying channel-wise dropout directly to the data demonstrates stable gains across both modalities.

STATEMENTS

Our work adheres to the ICLR Code of Ethics. Regarding the use of large language models (LLMs), we employed LLMs exclusively for writing polish. To ensure reproducibility, our code and the guidance of our code are available at `https://anonymous.4open.science/r/DRIP_ICLR26-C766`. The link is fully anonymous and contains no information that could identify the authors.

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

## A  DETAIL OF THEORETICAL ANALYSIS

### A.1  DISCUSSION OF EXAMPLE 1

**Example 1.** *(Pseudo-invariant Features) Suppose training domains include $\mathcal{E}_{tr} = \{1, 2\}$ and target domain includes $\mathcal{E} \backslash \mathcal{E}_{tr} = \{3\}$. The causal model is defined as:*

$$Z_{spu}^{p,e=1} = Z_{spu}^{s,e=1} \quad = Z_{inv}^{e=1} \sim Bernoulli(0.8)$$
$$Z_{spu}^{p,e=2} = 1 - Z_{spu}^{s,e=2} \quad = Z_{inv}^{e=2} \sim Bernoulli(0.2)$$
$$1 - Z_{spu}^{p,e=3} = Z_{spu}^{s,e=3} \quad = Z_{inv}^{e=3} \sim Bernoulli(0.8)$$
$$Z_{spu} = (Z_{spu}^p, Z_{spu}^s), \quad X = (Z_{inv}, Z_{spu}), \quad Y = Z_{inv} \oplus N$$

*where $N \sim Bernoulli(0.1)$ is a binary noise, $N \perp\!\!\!\perp (Z_{inv}, Z_{spu})$ and $\oplus$ is XOR operation.*

It is clear that $Y$ is correlated with $Z_{\text{inv}}$ for any $e \in E$. $Z_{\text{spu}}^{\text{p}}$ is correlated with $Y$ in $e = 1, 2$, but shows anti-correlation in $e = 3$. $Z_{\text{spu}}^{\text{s}}$ is correlated with $Y$ in $e = 1, 3$, but shows anti-correlation in $e = 2$. From the global perspective ($\mathcal{E} = \{1, 2, 3\}$), both $Z_{\text{spu}}^{\text{p}}$ and $Z_{\text{spu}}^{\text{s}}$ are spurious, since their relationship with $Y$ changes with $E$. However, if we only focus on $\mathcal{E}_{\text{tr}} = \{1, 2\}$, the relationship between $Z_{\text{spu}}^{\text{p}}$ and $Y$ appears invariant. In our example, this leads to $Z_{\text{spu}}^{\text{p}} = Z_{\text{inv}}$ in $\mathcal{E}_{\text{tr}}$. In more general real-world cases, this equivalence does not necessarily hold, but such pseudo-invariance is still widespread. In this situation, no matter how the regularization term is designed, it is impossible to distinguish $Z_{\text{spu}}^{\text{p}}$ from $Z_{\text{inv}}$ using only $\mathcal{E}_{\text{tr}}$. In contrast, the relationship between $Z_{\text{spu}}^{\text{s}}$ and $Y$ varies across $\mathcal{E}_{\text{tr}}$. An algorithm based on the invariance principle is therefore likely to detect it.

## A.2 Proof of Theorem 4

**Lemma 8.** $P(Y \mid Z_{inv}) = P(Y \mid R(X))$.

*Proof.* By Condition A of Definition 2, we have:

$$P(Y \mid X) = P(Y \mid R(X)).$$

By Assumption 3, $X$ is information-equivalent to $(Z_{\text{inv}}, Z_{\text{spu}}, N_x)$. By causal graph in Figure 1, the only path between $Y$ and $Z_{\text{spu}}$ is $Y \leftarrow Z_{\text{inv}} \leftarrow D \rightarrow Z_{\text{spu}}$. By d-separation, $Y \perp\!\!\!\perp Z_{\text{spu}} \mid Z_{\text{inv}}$. With $N_x$ being independent, we have

$$P(Y \mid X) = P(Y \mid Z_{\text{inv}}, Z_{\text{spu}}, N_x) = P(Y \mid Z_{\text{inv}})$$

This implies:

$$P(Y \mid R(X)) = P(Y \mid X) = P(Y \mid Z_{\text{inv}})$$

$\square$

**Lemma 9.** $H(Z_{inv} \mid R(X)) = 0$.

To prove the lemma, we require the following assumption: the causal relationship between $Z_{\text{inv}}$ and $Y$ is identifiable, i.e.,

**Assumption 10.** *(Identifiability Assumption)*

$$\forall z_1, z_2, z_1 \neq z_2, \quad \text{we have} \quad P(Y \mid Z_{\text{inv}} = z_1) \neq P(Y \mid Z_{\text{inv}} = z_2).$$

This is reasonable since $Z_{\text{inv}}$ is defined to capture the causal features of $Y$. If two values of $Z_{\text{inv}}$ induced the same distribution of $Y$, they could be merged without loss of predictive information, making $Z_{\text{inv}}$ a minimal sufficient representation of $Y$. Similar assumptions are common in existing works (Peters et al., 2016; Sun et al., 2020; Pearl, 2009). We note that this is a technical assumption needed for theoretical guarantees and may not always hold in practice, but it is justified here as $Z_{\text{inv}}$ is a hypothetical latent variable.

*Proof.* We argue by contradiction. Suppose $H(Z_{\text{inv}} \mid R(X)) > 0$. Then there must exist two distinct inputs $x_1 = c(z_1, s_1, n_1)$ and $x_2 = c(z_2, s_2, n_2)$ such that their invariant components differ, i.e., $z_1 \neq z_2$, but they are mapped by $R$ to the same representation $r$:

$$R(x_1) = R(x_2) = r.$$

From Lemma 8, for $x_1$ we have

$$P(Y \mid R(X) = r) = P(Y \mid Z_{\text{inv}} = z_1) = P(Y \mid Z_{\text{inv}} = z_2), \qquad z_1 \neq z_2,$$

which contradicts Assumption 10. $\square$

Then, we can prove:

**Theorem 4.** $I(R(X); Z_{inv}) = I(X; Z_{inv})$.

*Proof.* We now have $H(Z_{\text{inv}} \mid R(X)) = 0$, which implies that $Z_{\text{inv}}$ is measurable with respect to $R(X)$ (almost surely). Thus we have $Z_{\text{inv}} \perp\!\!\!\perp X \mid R(X)$ and then $I(Z_{\text{inv}}; X \mid R(X)) = 0$. Applying the chain rule of mutual information gives

$$I(Z_{\text{inv}}; X, R(X)) = I(Z_{\text{inv}}; R(X)) + I(Z_{\text{inv}}; X \mid R(X)) = I(Z_{\text{inv}}; R(X)).$$

Since $R(X)$ is a deterministic function of $X$, we have $I(Z_{\text{inv}}; X, R(X)) = I(Z_{\text{inv}}; X)$, which implies

$$I(X; Z_{\text{inv}}) = I(R(X); Z_{\text{inv}}).$$

$\square$

### A.3 PROOF OF THEOREM 5

**Theorem 5.** $I(R(X); Z_{spu}, N_x) < I(X; Z_{spu}, N_x)$.

*Proof.* By Condition B of Definition 2, we have $H(X) - H(R(X)) = H(X \mid R(X)) > 0$. We define $Z_u := (Z_{\text{spu}}, N_x)$. By Assumption 3, $X$ is information-equivalent to $(Z_{\text{inv}}, Z_{\text{spu}}, N_x)$, and then we have

$$H(X) - H(R(X)) = H(Z_{\text{inv}}, Z_u \mid R(X)).$$

By the chain rule of conditional entropy and Lemma 9, we have

$$H(Z_{\text{inv}}, Z_u \mid R(X)) = H(Z_{\text{inv}} \mid R(X)) + H(Z_u \mid Z_{\text{inv}}, R(X)) = H(Z_u \mid Z_{\text{inv}}, R(X)),$$

which implies

$$H(X) - H(R(X)) = H(Z_{\text{inv}}, Z_u \mid R(X)) = H(Z_u \mid Z_{\text{inv}}, R(X)).$$

We now turn to consider $I(X; Z_u) - I(R(X); Z_u)$. By the chain rule for mutual information and the definition of conditional mutual information, we have

$$I(X; Z_u) - I(R(X); Z_u) = I(X; Z_u \mid R(X)) = H(Z_u \mid R(X)) - H(Z_u \mid X, R(X)).$$

Since the information equivalence between $X$ and $(Z_{\text{inv}}, Z_u)$, the second term is zero: $H(Z_u \mid X, R(X)) = H(Z_u \mid X) = 0$. Then we have

$$I(X; Z_u) - I(R(X); Z_u) = H(Z_u \mid R(X))$$

Since conditioning cannot increase entropy:

$$H(Z_u \mid R(X)) \geq H(Z_u \mid R(X), Z_{\text{inv}})$$

Then we have

$$I(X; Z_u) - I(R(X); Z_u) \geq H(X) - H(R(X)) > 0$$

Which is equivalent to

$$I(X; Z_{\text{spu}}, N) - I(R(X); Z_{\text{spu}}, N_x) \geq H(X) - H(R(X)) > 0$$

$\square$

In addition to proving this theorem, we also show that the reduction effect of the operator $R$ on spurious features is no weaker than its compression effect on the data. This provides a quantitative estimate of the benefit of $R$.

### A.4 JUSTIFICATION FOR CRITERION 6

**Criterion 6** (Informal)**.** *If the final converged value of the training loss with $R$ is similar to that without $R$, we consider $R$ to satisfy Condition A.*

The justification for our criterion rests on the assumption of an ideal model with sufficient capacity to fully capture the correlation between $X$ (or $R(X)$) and $Y$. For such a model, the minimum achievable loss is determined by the inherent information content, as described by the conditional probability $P(Y \mid X)$. If a transformation $R$ preserves this information such that $P(Y \mid R(X)) = P(Y \mid X)$, then the ideal model's minimum possible loss must be identical

Table 4: Dataset details and dataset-specific hyper-parameters.

| Datasets | Input Channels | Classes | Users | Epochs | Learning Rate | Filter Channels |
|---|---|---|---|---|---|---|
| DSADS | 30 | 19 | 8 | 20 | 5e-4 | 64, 128, 256 |
| OPPORTUNITY | 42 | 18 | 4 | 150 | 5e-3 | 128, 256, 512 |
| PAMAP2 | 18 | 19 | 8 | 50 | 5e-4 | 128, 256, 512 |
| MOTIONSENSE | 6 | 6 | 24 | 150 | 5e-3 | 128, 256, 512 |
| HAPT | 6 | 12 | 30 | 100 | 1e-3 | 64, 128, 256, 512 |

for both $X$ and $R(X)$. Consequently, observing a similar converged loss between the two scenarios provides strong evidence that the transformation has not discarded label-relevant information, as any such loss would necessarily increase the optimal error rate.

To best approximate the conditions of this ideal assumption in practice, we implemented several key experimental designs. We ensured that the models were trained long enough for their loss to fully converge, and maintained strictly identical training configurations for both settings to enable a fair comparison. Furthermore, we conducted multiple trials for each experiment to mitigate the effects of stochasticity and ensure our results were robust. Designs above led to the training loss curve in Figure 2a.

## B  DETAILED SETUP OF IMU EXPERIMENTS

We adopt a ResNet-like CNN as backbone, and the models are trained from scratch. We use 3 for kernel size and 0.2 for dropout within the model. We use a 2-layer projection head with 128 for the hidden dimension. In each CNN layer, we adopt 1-d Batch Norm, 1-d Dropout, residual connection and an average pooling with 2 for kernel size. Pooling in the last layer is disabled. All datasets are resampled to 50Hz and processed by sliding windows with the window size of 2.56s and the overlap of 1.28s. We present dataset details and dataset-specific hyper-parameters in Table 4.

