# OpenReview forum: "DRIP: Invariance-preserving Data Reduction for Domain Generalization"
_ICLR.cc/2026/Conference — ICLR 2026 Conference Withdrawn Submission_

### Official Review · Reviewer_kbks · 2025-10-23

**Soundness:** 3
**Presentation:** 3
**Contribution:** 3
**Rating:** 6
**Confidence:** 2

**Summary:**

The paper introduces DRIP (a data transformation family that (A) preserves label-relevant information, $(P(Y|R(X))=P(Y|X))$, while (B) reducing input entropy $(H(R(X))<H(X))$. Under an invertible generative assumption for (X), the authors prove DRIP preserves mutual information with invariant (causal) features and lowers information tied to spurious ones (Theorems 4–5). A simple channel-wise input dropout is proposed as a practical DRIP instance, evaluated on DomainBed (vision) and multiple IMU HAR datasets. Results show consistent but modest average gains over strong baselines; patch/pixel-wise variants are weaker.

**Strengths:**

1. **Clear, principled objective**: Formalizes a compression-with-invariance lens for DG and ties it to causal desiderata (Theorems 4–5).
2. **Simple, low-cost implementation**: Channel-wise dropout is trivial to plug in and shows the best behavior among tested variants.
3. **Empirical coverage**: Improvements on DomainBed  and across several IMU datasets; analyses of dropout rate and #domains are helpful.

**Weaknesses:**

1. **Strong Condition A**: Requiring $(P(Y|R(X))=P(Y|X))$ is rarely verifiable; the paper’s empirical proxy (comparing converged training losses) is suggestive but not a test of conditional distributions or invariance across environments. This risks circular validation.
2. **Heavy modeling assumptions**: Invertibility of $(X=c(Z_{inv}, Z_{spu},N))$ and identifiability are strong; many real pipelines violate them (augmentations, compression, quantization). Theory may not transfer intact.
3. **Modest margins and sensitivity**: Gains on DomainBed are incremental on average; pixel/patch variants underperform in places, and channel-wise dropout can fully erase inputs (RGB), forcing ad-hoc “net-loss” adjustments downstream. More ablations are needed to rule out selection effects.
4. **Baselines scope**: Comparisons omit several shape/texture-robust augmentations (Cutout/Random Erasing/Hide-and-Seek variants tuned for DG) and recent DG regularizers; fairness/tuning parity across methods is unclear. (Results tables list many classics but not all relevant augmentation-centric DG lines.)
5. **Mechanistic evidence**: No causal/representation probes showing spurious-feature attenuation (background dependence tests, causal edit robustness, or invariance diagnostics per domain).

**Questions:**

Please see the weaknesses above.

---

### Official Review · Reviewer_BRJW · 2025-10-25

**Soundness:** 2
**Presentation:** 3
**Contribution:** 2
**Rating:** 4
**Confidence:** 4

**Summary:**

This paper introduces DRIP (Data Reduction with Invariance-Preserving), a theoretically grounded data augmentation paradigm for DG. The core idea is to apply a transformation R that: 	(A) preserves the label-relevant information: P(Y|R(X)) = P(Y|X), (B) reduces the overall entropy of the input: H(R(X)) < H(X).  The authors provide theoretical justification that such transformations reduce spurious correlations while preserving invariant features, and they propose several low-cost implementations (e.g., channel-wise dropout) for both image and IMU domains. Experimental results on DomainBed and IMU datasets show performance gains over prior DG baselines.

**Strengths:**

- Provides a clean, theoretically-motivated definition of a class of data augmentations (DRIP) for DG.
- Introduces a simple but effective implementation (channel-wise dropout), which slightly outperforms ERM and DG baselines on DomainBed and IMU datasets.
- Includes theoretical proofs (Theorems 4 and 5) and a discussion of pseudo-invariant feature leakage—a known challenge in DG.-
- Empirical evaluation is thorough, covering multiple datasets and modalities, with reasonable ablations.

**Weaknesses:**

1. Weak verification of DRIP conditions: The verification of Condition A relies solely on training loss, which cannot support the verification of the core condition A of DRIP in the document "P(Y｜R(X)=P(Y｜X)" ("Retain the necessary information for label prediction", formalized as P(Y｜X)=P(Y｜R(X))". It is completely dependent on Criterion 6: "If a sufficiently strong model, after applying R, the convergence value of training loss is similar to that without R, then R satisfies Condition A." This verification logic has certain defects, and the document does not supplement more rigorous verification methods.
2. The core implementation of DRIP proposed in the document is essentially a slight adjustment to the early input-level dropout method, without any breakthrough innovation, and the document itself also acknowledges the similarity to existing methods.
3. The two core assumptions (Assumption 3, Assumption 10) supporting the DRIP theory both have the problem of being "overly idealized," and the document does not prove their validity in the experimental data used.
4.The document proves that DRIP "retains unchanged features and reduces false features" through mutual information (I(X; Z_inw), I(X; Z_spu)), but it has not established a mathematical connection between this conclusion and "reduced generalization error," leading to a disconnect between theoretical analysis and experimental results.

**Questions:**

1. Can a more direct method be provided to verify condition A? Can mutual information or conditional entropy estimation be tried instead of relying solely on "loss values"? Have you considered estimation under the information bottleneck objective?
2. dropout specific modus operandi needs to be refined -Dropout is random for each sample, or fixed mask? Does it consider whether the training labels are still separable after masking?
3. Is DRIP applicable to other modalities such as language, text, and graph neural networks?
4. Does DRIP still hold when label noise exists? The current assumption is that the data is clean, but in reality, noise is quite common.

---

### Official Review · Reviewer_kEjn · 2025-10-31

**Soundness:** 1
**Presentation:** 2
**Contribution:** 1
**Rating:** 2
**Confidence:** 5

**Summary:**

This paper proposes DRIP, a theoretically-motivated data augmentation paradigm for domain generalization (DG). DRIP identifies data transformations that compress the input representation while strictly preserving label-relevant information, aiming to remove spurious features while retaining invariant (causal) ones. The authors establish rigorous conditions under which such operators theoretically improve DG, provide formal proofs, and propose practical simple implementations—especially channel-wise dropout. The effectiveness of DRIP is validated empirically on standard visual and IMU-based HAR benchmarks, showing improvements over various strong DG baselines.

**Strengths:**

1.	The paper proposes a theoretical concept of DRIP for causality-based DG and proposes multiple implementations of DRIP.
2.	Experiments on multiple DG datasets prove the effectiveness of the method.
3.	The paper is well-organized.

**Weaknesses:**

1.	The motivation is redundant. The authors claim that their motivation is to provide theoretical analysis for causality-based methods, which has already been extensively studied [1, 2, 3]. Moreover, the theoretical result for augmentation, i.e., “compresses information while preserving the information necessary for classification”, is essentially indistinguishable from existing causal-learning formulations. Its conclusions also overlap with prior domain-specific feature suppression methods. The manuscript neither identifies nor addresses a new problem.
2.	The contributions are weak.The theoretical results do not present original or substantive advances. In addition, the implementation of DRIP is, in essence, random channel-wise dropout, whose effectiveness has been established in prior work [4, 5, 6]. The multiple DRIP implementations offer no clear novelty. Finally, the claim to be “the first theoretically grounded data augmentation that demonstrates effectiveness across multiple modalities” is untenable. There already exist general theoretical treatments of data augmentation that apply across modalities—even when not stated explicitly. For instance, Mixup has demonstrated effectiveness in language, vision, EEG, and other modalities.
3.	The experiments are limited. The multiple DRIP implementations (which are actually random dropouts) deliver slight gains and are not compared against recent DG methods [7,8]. The connection between the analytical experiments and the theory is weak. The experiments prove the known effectiveness of random channel dropout, a point that has already been supported both theoretically and empirically in prior studies [5,6].

[1] Lv, Fangrui, et al. "Causality inspired representation learning for domain generalization." Proceedings of the IEEE/CVF conference on computer vision and pattern recognition. 2022.

[2] Mahajan, Divyat, Shruti Tople, and Amit Sharma. "Domain generalization using causal matching." International conference on machine learning. PMLR, 2021.

[3] Mo, Zhenling, Zijun Zhang, and Kwok-Leung Tsui. "Domain Generalization Study of Empirical Risk Minimization from Causal Perspectives." IEEE Transactions on Multimedia (2025).

[4] Park, Sungheon, and Nojun Kwak. "Analysis on the dropout effect in convolutional neural networks." Asian conference on computer vision. Cham: Springer International Publishing, 2016.

[5] Srivastava, Nitish, et al. "Dropout: a simple way to prevent neural networks from overfitting." The journal of machine learning research 15.1 (2014): 1929-1958.

[6] Guo, Jintao, et al. "PLACE dropout: A progressive layer-wise and channel-wise dropout for domain generalization." ACM Transactions on Multimedia Computing, Communications and Applications 20.3 (2023): 1-23.

[7] Wen, Changsong, et al. "Domain Generalization in CLIP via Learning with Diverse Text Prompts." Proceedings of the Computer Vision and Pattern Recognition Conference. 2025.

[8] Wang, Shanshan, et al. "Exploring Invariance Matters for Domain Generalization." IEEE Transactions on Image Processing (2025).

**Questions:**

See in the Weaknesses.

---

### Official Review · Reviewer_C3aH · 2025-11-01

**Soundness:** 2
**Presentation:** 2
**Contribution:** 2
**Rating:** 4
**Confidence:** 4

**Summary:**

This paper proposes DRIP (Data Reduction with Invariance-Preserving), a simple method for Domain Generalization (DG) that aims to preserve label-relevant information while reducing spurious domain-dependent information.
The authors formalize an information-theoretic definition of a transformation ( R(X) ) that satisfies two conditions:

$
P(Y|R(X)) = P(Y|X), \quad H(R(X)) < H(X)
$

—that is, (R(X)) reduces input information entropy while preserving label semantics.
They prove two theorems: (1) DRIP preserves invariant information, and (2) DRIP reduces spurious information.
In practice, DRIP is implemented through channel-wise dropout (randomly dropping RGB or sensor channels) before standard ERM training. Experiments on vision and IMU datasets show moderate gains over baselines like IRM and IIB, suggesting DRIP is a simple yet effective regularization that enhances domain robustness.

**Strengths:**

* **Clear theoretical framing of “safe” data augmentation.**
  The paper introduces a simple information-theoretic rule — (P(Y|R(X)) = P(Y|X)) and (H(R(X)) < H(X)) — to describe when an augmentation is safe, meaning it keeps label semantics intact while removing redundant input information.

* **Straightforward and broadly applicable idea.**
  DRIP can use any compression function, deterministic or random, making it easy to plug into ordinary ERM training without extra modules or loss terms.

* **Evidence across different data types.**
  The method works on both images and sensor signals, giving consistent (though not dramatic) gains. This supports that the approach is lightweight yet somewhat robust across modalities.

**Weaknesses:**

1. **Implementation overlaps with common data augmentation.**
   The core implementation—channel-wise `Dropout2d` on RGB inputs—is effectively a data augmentation technique already used in regularization. The main novelty lies in its theoretical framing, not the algorithm itself. While the authors claim conceptual distinction, in practice it behaves much like input-level dropout or Cutout. The paper would benefit from more evidence that DRIP behaves differently from standard augmentations beyond small accuracy gains.

2. **Limited causal and semantic reach.**
   Dropping RGB channels may reduce domain bias from color and texture, but doesn’t address higher-level spurious correlations like object–background co-occurrence (e.g., camel–desert, cow–grass). The method primarily weakens low-level style bias and does not modify semantic dependencies. Hence, the causal interpretation—removing spurious features—is overstated relative to what channel masking can achieve.

3. **Weak experimental evidence for claimed mechanism.**
   The experiments show modest accuracy improvements against (1–2pp) across standard DomainBed datasets (no comparison to domain randomization methods, which are augmentation-wise similar to this paper ), without statistical significance or probing of learned features. There is no qualitative or representational analysis—no visualization or domain alignment metrics—to demonstrate that DRIP truly suppresses spurious correlations. As a result, the empirical section supports it as a lightweight regularizer but not as a verified causal-invariance method.

**Questions:**

- Would hidden-layer dropout (on features ) satisfy similar theoretical guarantees? If not, what prevents the framework from being extended beyond input-level transformations?

- Since DRIP only removes low-level (color/texture) bias, have the authors tested its effectiveness on datasets where domain shift is semantic rather than stylistic?

---

### Note · Authors · 2025-11-23

I have read and agree with the venue's withdrawal policy on behalf of myself and my co-authors.